Diversity and spoilage potential of microbial communities associated with grape sour rot in eastern coastal areas of China

Gao Huanhuan gaohuanhuan368@126.com 1 2
Yin Xiangtian 1
Jiang Xilong 1
Shi Hongmei 1
Yang Yang 1
Wang Chaoping 1
Dai Xiaoyan 1 2
Chen Yingchun 1
Wu Xinying echomoon0622@163.com 1
1 Shandong Academy of Grape , Jinan , China
2 Shandong Academy of Agricultural Sciences, Institute of Plant Protection , Jinan , China
LaMontagne Michael
Electronic publication date: 2020 Jun 16
Publication date: 2020
Volume: 8
Electronic Location ID: e9376
Received 2019 May 31; Accepted 2020 May 27
Copyright: ©2020 Gao et al.
Copyright year: 2020
Copyright holder: Gao et al.
License: This is an open access article distributed under the terms of the Creative Commons Attribution License, which permits unrestricted use, distribution, reproduction and adaptation in any medium and for any purpose provided that it is properly attributed. For attribution, the original author(s), title, publication source (PeerJ) and either DOI or URL of the article must be cited.
License URL: https://creativecommons.org/licenses/by/4.0/

Keywords: Grape, High-throughput sequencing, Bacteria, Pathogenicity, Fungus

Funding: Natural Science Foundation of China No. 31801750 Fruit innovation team of Shandong modern agricultural industry technology system No. SDAIT-06-21 Agricultural scientific and technological innovation project of Shandong Academy of Agricultural Sciences No. CXGC2018E17 This work was supported financially through a grant from the Natural Science Foundation of China (No. 31801750), Fruit innovation team of Shandong modern agricultural industry technology system (No. SDAIT-06-21) and Agricultural scientific and technological innovation project of Shandong Academy of Agricultural Sciences (No. CXGC2018E17). The funders had no role in study design, data collection and analysis, decision to publish, or preparation of the manuscript.

==============================
As a polymicrobial disease, sour rot decreases grape berry yield and wine quality. The diversity of microbial communities in sour rot-affected grapes depends on the cultivation site, but the microbes responsible for this disease in eastern coastal China, has not been reported. To identify the microbes that cause sour grape rot in this important grape-producing region, the diversity and abundance of bacteria and fungi were assessed by metagenomic analysis and cultivation-dependent techniques. A total of 15 bacteria and 10 fungi were isolated from sour rot-affected grapes. High-throughput sequencing of PCR-amplicons generated from diseased grapes revealed 1343 OTUs of bacteria and 1038 OTUs of fungi. Proteobacteria and Firmicutes were dominant phyla among the 19 bacterial phyla identified. Ascomycota was the dominant fungal phylum and the fungi Issatchenkia terricola, Colletotrichum viniferum, Hanseniaspora vineae, Saprochaete gigas, and Candida diversa represented the vast majority ofmicrobial species associated with sour rot-affected grapes. An in vitro spoilage assay confirmed that four of the isolated bacteria strains (two Cronobacter species, Serratia marcescens and Lysinibacillus fusiformis) and five of the isolated fungi strains (three Aspergillus species, Alternaria tenuissima, and Fusarium proliferatum) spoiled grapes. These microorganisms, which appear responsible for spoiling grapes in eastern China, appear closely related to microbes that cause this plant disease around the world.

Introduction

Grape sour rot is a polymicrobial disease characterized by the disaggregation of the internal tissues of berries, detachment of the rotten berry from the pedicel, and a strong ethyl acetate smell. This disease causes millions of dollars in revenue loss each year due to decreases in the quality of berries (Barata et al., 2011; Steel, Blackman & Schmidtke, 2013). A number of microorganisms, such as Ascomycota, acetic acid bacteria (AAB), and filamentous fungi, can infect ripe and thin-skinned grape berries (Nally et al., 2013). Acetic acid released by AAB attracts the fruit fly Drosophila, which contributes to sour rot (Hall et al., 2018). The composition of microorganisms in sour rot-affected grapes depends on the cultivation site and grape variety.

The frequency and density of yeast species associated with sour rot differ between grape cultivars. The most frequently recovered ascomycetous species from rotten wine grapes are Candida krusei, Kloeckera apiculata, and Metschnikowia pulcherrima and a less frequent species is Issatchenkia occidentalis (Guerzoni & Marchetti, 1987). Barata et al. (2008) reported that Candida vanderwaltii, Hanseniaspora uvarum, and Zygoascus hellenicus are the most frequent species in rotten Trincadeira Preta red grape. The relative abundance of these microorganisms depends on the ripening stage and the availability of nutrients. Basidiomycetes and the yeast-like fungus Aureobasidium pullulans dominate intact grape berries. Ascomycetes with higher fermentative activity, like Pichia spp., Zygoascus hellenicus, wine spoilage yeasts, and AAB, are more frequent in rotten grape samples than in healthy grapes (Barata, Malfeito-Ferreira & Loureiro, 2012a; Barata, Malfeito-Ferreira & Loureiro, 2012b). Other than the widespread Hanseniaspora uvarum in sour rot wine grapes and table grapes, non-saccharomyces yeast (NSY) and AAB species occur in sour rot table grape. Pinto et al. (2017) proved that among all NSY-AAB associations, the yeastbacterium association composed of Candida zemplinina CBS 9494 and Acetobacter syzygii LMG 21419 shows the highest prevalence. This microbial consortium produces spoilage metabolites such as acetic acid and gluconic acid (Pinto et al., 2019).

Advances in molecular biology techniques and metagenomics have facilitated microbial community analyses (Andreote, Azevedo & Araújo, 2009) and characterization of microbes associated with plant diseases (Huang et al., 2017; Shen et al., 2018; Brady et al., 2017). Hall, O’Bryon & Osier (2019) characterized the microbiome of sour rot-affected grapes in New York by high-throughput sequencing and found that Acetobacter species were significantly more abundant in symptomatic samples than in asymptomatic ones. Studies of the microbial diversity of bacteria and fungi in rotten grapes in the eastern coast of China, a very important grape growing region, are limited.

In this study, metagenomic analysis was used to determine the diversity and abundance of bacteria and fungi in sour rot-affected table grapes collected from Yantai city. In parallel, we isolated several microbes and determined their potential to spoil grapes in an in vitro assay.

Materials & Methods

Samples of sour rot-affected grapes

Sour rot-affected grapes infested with fruit flies were collected from vineyards in Yantai (N36°27′, E117°10′), Shandong Province, China. Approximately 1.0 g of rotten tissue was sliced from each sour rot-affected grape (Muscat), and the tissues from 100 sour rot-affected grapes were collected together into one 50-mL sterile centrifuge tube. Three replicates from a total of 300 sour rot-affected grapes were stored at −80 °C for 16S rDNA and ITS high-throughput sequencing. Another three replicates were used for the separation and identification of culturable microorganisms in sour rot-affected grapes.

16S rDNA and ITS high-throughput sequencing analysis

DNA extraction and Illumina MiSeq sequencing of 16S rDNA and ITS genes

DNA was extracted from three rotten group samples using the Insect DNA Kit (OMEGA) and further purified using the MoBio PowerSoil Kit. The bacterial universal primers 341 F (5′-CCTACACGACGCTCTTCCGATCTN (barcode) CCTACGG-GNGGCWGCAG-3′) and 805 R (5′-GACTGGAGTTCCTTGGCACCCGAGAA- -TTCCA (barcode) GACTACHVGGGTATCTAATCC-3′) were used for amplification of the V3–V4 region of 16S rDNA. The fungal universal primers ITS4 F (5′-CCCTACACGACGCTCTTCCGATCTN (barcode) TCCTCCGCTTATTGATATG-3′) and ITS3 R (5′-GTGACTGGAGTTCCTTGG CACCCGAGAATTCCAGCATCGAT- -GAAGAACG–CAGC-3′) were used for amplification of the ITS gene. Each reaction comprised 15 µL of Phurs Mix (2er), 1.5 µL of each primer, 10 ng of template DNA, and ddH2O. The cycling conditions were as follows: initial denaturation at 95 °C for 1 min, followed by 35 cycles at 95 °C for 10 s, 54 °C for 30 s, and 72 °C for 30 s, and a final extension at 72 °C for 5 min. ITS was amplified under similar cycling conditions, except for the annealing temperature (52 °C). The sequencing libraries for 16S rDNA and ITS were constructed using the TruSeq DNA PCR-Free Preparation Kit (Illumina, San Diego, CA, USA) and quantified using Qubit 3.0 (Life Technologies, Grand Island, NY, USA). Then, the library was sequenced on an Illumina MiSeq platform (HiSeq 2000; PE250). After the removal of low-quality reads and primer/adaptor sequences using SeqClean, high-quality reads (clean data) were generated and used for further analysis. These sequencing procedures were performed by Sangon Biotech (Shanghai, China) Co., Ltd.

Alpha diversity analysis

Sequences were clustered into operational taxonomic units (OTUs) using the 97% identity threshold (3% dissimilarity level) (Mothur, https://mothur.org/) (Schloss et al., 2009). The reads of 16S rDNA and ITS sequences had been submitted in the Short Read Archive (BioProject ID: PRJNA61015). According to the number of OTUs, the rarefaction curve for 16S rDNA and ITS sequences was made to measure the adequacy and rationality of data. Shannon and Simpson diversity indexes were calculated as indicators of microbial diversity, and Chao1 and ACE indices were calculated as indicators of microbial richness using Mothur (Schloss et al., 2009). All OTUs were analyzed using BLASTN and the 16S rDNA and ITS databases (http://ncbi.nlm.nih.gov/). The best results (similarity >90% and coverage>90%) were used for subsequent classification. The sequences that did not satisfy these criteria were defined as “unclassified.” Species richness and relative abundances were estimated.

Diversity of culturable microorganisms in sour rot-affected grapes

The above samples with three replicates were suspended in phosphate-buffered saline (PBS, 0.2 M, pH 7.2) at ratios of 1:103, 1:104 and 1:105. The suspension (200 µL; different concentrations) was spread on nutrient agar medium and potato dextrose agar medium with three replicates each. After culturing at 25 °C for 48 h on nutrient agar and 7 d on potato dextrose agar a total of 80 colonies were picked and restreaked for purity using the primary media.

Identification of cultivated bacteria

Physiological and biochemical characteristics of each bacterium were analyzed according to the methods described by Dong & Cai (2001). The following tests were performed: Gram staining, spore staining, bacterial motility test, catalase reaction, methyl red test, starch hydrolysis, benzopyrrole test, VP test, malonic acid test, gelatin test, H2S test, citrate test, ammonia production test, litmus milk test, and urease test.

DNA was extracted from a single colony of each bacterium using the Bacterial DNA Kit (OMEGA, Norcross, GA, USA) and purified using the DNA Clean-Up Kit (OMEGA). 16S rDNA was amplified for each DNA template using the Bio-Rad 1000-Series Thermal Cycler PCR (Hercules, CA, USA). The thermal cycling profile was as follows: initial denaturation at 95 °C for 3 min, followed by 35 cycles of 95 °C for 15 s, 54 °C for 30 s, and 72 °C for 1 min, and a final extension at 72 °C for 5 min. Primer sequences were as follows: 16S rDNA-27F: 5′-AGAGT TTGATCCTGGCTCAG-3′; 16S rDNA-1492R: 5′-TACGGYTACCTTGTTACGACTT-3′.

Identification of cultivated fungi

The morphological features of each fungus were analyzed using a light microscope (CX41RF; Olympus, Tokyo, Japan) according to the methods described by Dai (1978). The mycelium of each purified fungus was collected in PDA medium. DNA was extracted using the Fungal DNA Kit (OMEGA) and purified using the DNA Clean-Up Kit (OMEGA). The ITS gene was amplified according to the following thermal cycling profile: initial denaturation at 95 °C for 3 min, quantification for 35 cycles (95 °C for 15 s followed by 52 °C for 30 s and 72 °C for 1 min), and a final extension at 72 °C for 5 min. Sequences of the universal primers were as follows: ITS1: 5′-TCCGTAGGTGAACCTGCGG-3′; ITS4: 5′-TCCTCCGCTTATTGATATGC-3′.

Sequencing

PCR products were purified using the TaKaRa Mini BEST Agarose Gel DNA Extraction Kit (Takara, Japan) and sequenced on an ABI-3730 DNA analyzer (Applied Biosystems, Foster City, CA, USA).The sequences were analyzed using BLAST (http://ncbi.nlm.nih.gov/). Phylogenetic trees of bacteria and fungi were separately constructed using the neighbor-joining method (NJ; Saitou & Nei, 1987) implemented in MEGA 6.0 (LynnonBiosoft, San Ramon, CA, USA). The sequences of bacteria and fungi were submitted to GenBank using SEQUIN (see phylogenetic trees for accession numbers).

Spoilage potential assay of cultivated bacteria and fungi

Isolated bacteria and fungi were tested for spoilage potential on grape berries. Healthy grape berries of Midnight Beauty, a susceptible variety, were collected and surface sterilized with 1% sodium hypochlorite (NaClO) solution for one minute. Excess NaClO was removed by washing the berries twice in sterile distilled water. The experimental berries were pricked 2–3 mm deep using a dissecting needle to simulate the wounds made by fruit flies during egg laying or other mechanical damage. The bacterial suspension and fungal spore suspension were prepared with a concentration of approximately 1 × 106 cfu/ml or conidia/ml in the suspension. This suspension (5 µl per berry) was used to inoculate the wounds of healthy grape berries. Sterile water was used instead of the suspension as a negative control. Two methods were used to observe the spoilage potential of bacteria and fungi. In the merged placement method, 10 grape berries were placed in a single Petri dish (10 cm in diameter and three cm in height) to simulate grape clusters in the field. In the separate placement method, each of 10 grape berries was placed in a single culture bottle (2.5 cm in diameter and three cm in height). Three replicates were established for both methods and each bacterium and fungus treatment. Subsequently, the inoculated grape berries were kept in a moisture chamber at 27 °C/25 °C (day/night) and 80% humidity, and symptoms were recorded on the 5th day. The bacterial and fungal species were reisolated from these artificially inoculated grape berries using NA medium and PDA medium, respectively. The resulting culture was compared with the original culture (Hyun et al., 2001).

Based on the ratio of the infected area to the total area, grading was performed as follows (Rouxel et al., 2013; Zhou et al., 2014): 0, no disease spot; 1, less than 5% of the total area infected; 3, 5% to 25% of the total area infected; 5, 25% to 50% of the total area infected; 7, 50% to 75% of the total area infected; 9, 75% to 100% of the total area infected. The incidence (%) and the McKinney index of bacteria and fungi were calculated according to the following formulas:

(1) Thepercentageofincidence%=100∗thenumberofdiseasedberriesthenumberofallberries

(2) TheMcKinneyindex=100∗∑k=0nkx9N

where, x is the value for each grade; n is the number of diseased berries at each level; and N is the total number of fruits investigated.

Results

Sequencing and alpha diversity analyses

The mean lengths of 16S rDNA of bacteria and ITS gene of fungi generated from metagenomic DNA extracted from rotting grapes were 413 ± 3 bp and 279 ± 5 bp, respectively. The OTU numbers for bacteria and fungi were 1,343 ± 283 and 1,039 ± 387 respectively (Table 1). Rarefaction curves of three samples for 16S rDNA and ITS sequences reached asymptotes (Fig. S1), which suggests coverage was sufficient. The number of sequences for each OTU decreased rapidly by the OTU rank of 18 (Fig. S1). The flat curve indicated a high degree of sequencing uniformity.

Table 1 Sequence information of bacterium and fungi in sour rot-affected grapes.

Group	Sample	Number of raw reads	Mean length of raw reads	Number of clean reads	Mean length of clean reads	Number of filtered reads	
16S rDNA	1	56,220	447	54,080	410	38,313	
2	63,599	458	61,916	418	39,939	
3	52,692	449	51,690	410	24,500	
Mean ± SE	57,504 ± 3,213	451 ± 3	55,895 ± 3,088	413 ± 3		
ITS	1	80,740	317.03	80,628	2,746	79,658	
2	71,362	332	71,281	2,886	71,160	
3	76,531	318	76,432	276	76,154	
Mean ± SE	76,211 ± 2,712	322 ± 5	76,114 ± 2,704	2,794 ± 5		

In a phylogenetic tree of the top 50 bacterial OTUs, 15 OTUs were classified as phylum Firmicutes, class Bacilli. Among the other 35 Proteobacteria, 21 OTUs belonged to the class Alphaproteobacteria, three to Betaproteobacteria, and 11 to Gammaproteobacteria (Fig. S2). In a phylogenetic tree of the top 50 fungal OTUs, one belonged to Basidiomycota and eight were not identified based on searches against the ITS database. Among the other 41 Ascomycota OTUs, 29 belonged to the class Saccharomycetes, nine to Sordariomycetes, and two to Dothideomycetes (Fig. S3).

The microbial diversity, as determined by the Shannon index and Simpson index, was higher for bacteria than fungi, and richness, as determined by the Chao1 index and ACE index, was higher for fungi than bacteria (Table 2).

Table 2 Diversity indices of bacterium and fungi in sour rot-affected grapes.

Parameters	Parameters	Bacterium (Mean ± SE)	Fungi (Mean ± SE)	
Diversity indices	Shannon	3 ± 1.3 E–01	220 E–2 ± 1.7 E–01	
ACE	22,034 ± 2,927	32,667 ± 1385	
Chao1	9,745 ± 1,430	10,779 ± 1476	
Simpson	1 E–01 ± 2 E–02	21 E-2 ± 4 E–02	
OTUs number	1,343 ± 283	1,039 ± 386	

Microbial taxonomic analysis

Proteobacteria (72%) and Firmicutes (27%) were dominant among the 19 bacteria phyla identified (Fig. 1A). The proportion of other bacteria was less than 1%. The dominant genera in sour rot-affected grapes were Acetobacter (38%), Gluconobacter (24%), Bacillus (12%), and Lactococcus (Fig. 1B).

Figure 1 The bacterial community structure in sour rot-infected grapes based on 16S rDNA high-throughput sequencing.

(A) The bacterial community structure based on genus; (B) the bacterial community structure based on phylum.

Ascomycota (94%) was the dominant phylum in the identified fungal community (Fig. 2A). The dominant species identified in sour rot-affected grapes were Issatchenkia terricola (18%), Colletotrichum viniferum (13%), Hanseniaspora vineae (13%), Saprochaete gigas (4%), and Candida diversa (4%), and 32% of isolates were not taxonomically identified (Incertae sedis sp.) (Fig. 2B).

Figure 2 The fungal community structure in sour rot-affected grapes based on ITS high-throughput sequencing.

(A) The fungal community structure based on genus; (B) the fungal community structure based on phylum.

Diversity of culturable microorganisms in sour rot-affected grapes

We cultured 15 bacterial strains from sour rot-affected grapes infested by fruit flies (Table 3). We identified Firmicutes as the dominant phylum (60%), with nine Gram-positive bacteria species (i.e., Staphylococcus saprophyticus, Lactococcus garvieae, Lactobacillus plantarum, two Lysinibacillus species, and four Bacillus species). We also isolated six Gram-negative bacteria species assigned to the phylum Proteobacteria. All bacterial taxa presented positive results in catalase reaction, gelatin, H2S, and ammonia production assays whereas they presented negative results for fermentation with the methyl red. Strains classified as Cronobacter malonaticus, Cronobacter sakazakii, and Klebsiella pneumoniae presented negative results for these biochemical tests (Table 4).

Table 3 Phylogeny of microbes isolated from sour rot-affected grapes.

Microorganism	Phylum	Species	Strain IDs	Accession numbers	
Bacterium	Proteobacteria	Cronobacter malonaticus	SRG1	MK743990	
Cronobacter sakazakii	SRG2	MK743989	
Klebsiella pneumoniae	SRG3	MK743987	
Acetobacter sp.	SRG4	MK743980	
Serratia marcescens	SRG5	MK743984	
Enterobacter hormaechei	SRG6	MK743988	
Firmicutes	Staphylococcus saprophyticus	SRG7	MK743982	
Lactococcus garvieae	SRG8	MK743983	
Lactobacillus plantarum	SRG9	MK743986	
Lysinibacillus fusiformis	SRG10	MK753026	
Lysinibacillus sp.	SRG11	MK743985	
Bacillus amyloliquefaciens	SRG12	MK743994	
Bacillus cereus	SRG13	MK743993	
Bacillus sp.- 1	SRG14	MK743992	
Bacillus sp.- 2	SRG15	MK743991	
Fungus	Deuteromycotina	Cladosporium oxysporum	SRG16	MK748311	
Alternaria tenuissima	SRG17	MK748314	
Saprochaete gigas or Geotrichum gigas	SRG18	MN567950	
Fusarium proliferatum	SRG19	MK748309	
Nigrospora sp.	SRG20	MK748317	
Ascomycotina	Penicillium citrinum	SRG21	MK748316	
Penicillium georgiense	SRG22	MK748315	
Aspergillus niger	SRG23	MK748313	
Aspergillus oryzae	SRG24	MK748312	
Aspergillus aculeatus	SRG25	MK748310	

Table 4 The physiological and biochemical characteristic of bacterium in sour rot-affected grape.

Bacterium	Strain IDs	Gram staining	Spore staining	Bacterial motility	Catalase reaction	Methyl red test	Starch hydrolysis test	Benzpyrole test	V-P test	
Cronobacter malonaticus	SRG1	–		–	+	–	–	–	+	
Cronobacter sakazakii	SRG2	–		–	+	–	–	–	+	
Klebsiella pneumoniae	SRG3	–		–	+	–	–	–	+	
Acetobacter sp.	SRG4	–		+	+	–	–	–	+	
Serratia marcescens	SRG5	–		+	+	–	+	–	+	
Enterobacter hormaechei	SRG6	–		+	+	–	+	–	+	
Staphylococcus saprophyticus	SRG7	+		+	+	–	+	–	+	
Lactococcus garvieae	SRG8	+		–	+	–	+	–	+	
Lactobacillus plantarum	SRG9	+		–	+	–	+	–	+	
Lysinibacillus fusiformis	SRG10	+	purple	+	+	–	+	–	–	
Lysinibacillus sp.	SRG11	+	purple	+	+	–	+	–	+	
Bacillus amyloliquefaciens	SRG12	+	pink	+	+	–	+	–	–	
Bacillus cereus	SRG13	+	purple	+	+	–	+	–	+	
Bacillus sp.-1	SRG14		purple	+	+	–	+	–	+	
Bacillus sp.-2	SRG15	+	purple	+	+	–	+	–	+	
Bacterium		Malonic acid test	Gelatin test	H2S test	Citrate test	Ammonia production test	Litmus milk test	Urease test	
Cronobacter malonaticus	SRG1	–	+	+	+	+	+	–	
Cronobacter sakazakii	SRG2	+	+	+	–	+	+	–	
Klebsiella pneumoniae	SRG3	+	+	+	+	+	–	+	
Acetobacter sp.	SRG4	+	+	+	+	+	+	+	
Serratia marcescens	SRG5	+	+	+	–	+	+	–	
Enterobacter hormaechei	SRG6	+	+	+	+	+	+	–	
Staphylococcus saprophyticus	SRG7	–	+	+	+	+	+	–	
Lactococcus garvieae	SRG8	–	+	+	–	+	+	–	
Lactobacillus plantarum	SRG9	+	+	+	+	+	+	–	
Lysinibacillus fusiformis	SRG10	+	+	+	–	+	–	–	
Lysinibacillus sp.	SRG11	+	+	+	+	+	–	–	
Bacillus amyloliquefaciens	SRG12	–	+	+	+	+	+	+	
Bacillus cereus	SRG13	–	+	+	+	+	+	–	
Bacillus sp.-1	SRG14	+	+	+	+	+	+	+	
Bacillus sp.-2	SRG15	–	+	+	+	+	+	+	

We cultured ten fungi from sour rot-affected grapes. Five were classified as Deuteromycotina, including Cladosporium oxysporum, Alternaria tenuissima, Geotrichum gigas, Fusarium proliferatum, and Nigrospora sp. (Table 3). C. oxysporum, with bottle-green colonies, developed into conidia by asexual reproduction. A. tenuissima colonies, with a white front side and brown reverse side, developed into conidia in the form of a chain lattice. The hyphae of Saprochaete gigas or Geotrichum gigas, with white colonies, developed into arthrospores by asexual reproduction. F. proliferatum, with red colonies, had branched conidiophores and sickle or long column-shaped conidia. Nigrospora sp. had irregular colonies, branched conidiophores, and ball-shaped conidia. Five species (i.e., Penicillium citrinum, P. georgiense, Aspergillus niger, A. aculeatus, and A. oryzae) belonged to Ascomycotina. The sporophores of P. citrinum and P. georgiense grew from hyphae and developed into brush-like structures. These two Penicillium species differed in colony color. The conidia of A. niger, A. aculeatus, and A. oryzae were black, green, and yellow, respectively (Fig. 3).

Figure 3 Colony morphology and the light mophology of the fungi in sour rot-affected grapes.

(A–E) represent the reverse side of colony morphology of Cladosporium oxysporum , Penicillium citrinum, Alternaria tenuissima, Saprochaete gigas, Fusarium proliferatum; (F–J) represent the front side of colony morphology of Cladosporium oxysporum, Penicillium citrinum, Alternaria tenuissima, Saprochaete gigas, Fusarium proliferatum; (K–O) represent the reverse side of colony morphology of P. georgiense, Aspergillus niger, Nigrospora sp., A. oryzae, A. aculeatus; (P–T) represent the font side of colony morphology of P. georgiense, Aspergillus niger, Nigrospora sp., A. oryzae, A. aculeatus. (U–DD) represent the light morphology of Cladosporium oxysporum, Alternaria tenuissima, Saprochaete gigas, Fusarium proliferatum, Nigrospora sp., Penicillium citrinum, P. georgiense, Aspergillus niger, A. oryzae, A. aculeatus.

Spoilage potential of culturable bacteria and fungi for grape sour rot

All 15 bacterial species and 10 fungal species demonstrated the potential to spoil grapes. In the merged placement method, all of the microorganisms except for B. amyloliquefaciens, caused cracking and infection in grapes (Fig. 4A). In the separate placement method, all of fungi except for Saprochaete gigas, could cause infection in grapes. Obvious symptoms were not detected in the grapes treated by bacterium (Fig. 4B). The bacterial species and the fungal species reisolated from these spoiled grapes using NA medium and PDA medium were confirmed as the original microorganisms summarized in Table 3.

Figure 4 The pathogenicity of bacteria and fungi in healthy grape berries.

(A–AA) represent pathogenicity of sterile water, LB medium, Cladosporium oxysporum, Alternaria tenuissima, Saprochaete gigas, Fusarium proliferatum, Nigrospora sp., Penicillium citrinum, P. georgiense, Aspergillus niger, A. oryzae, A. aculeatus, Cronobacter malonaticus, C. sakazakii, Klebsiella pneumoniae, Acetobacter sp., Serratia marcescens, Enterobacter hormaechei, Staphylococcus saprophyticus, Lactococcus garvieae, Lactobacillus plantarum, Lysinibacillus fusiformis, Lysinibacillus sp., Bacillus amyloliquefaciens, B. cereus, Bacillus sp.- 1, Bacillus sp.- 2 using the merged method; (BB–BBB) represent pathogenicity of sterile water, LB medium, Cladosporium oxysporum, Alternaria tenuissima, Saprochaete gigas, Fusarium proliferatum, Nigrospora sp., Penicillium citrinum, P. georgiense, Aspergillus niger, A. oryzae, A. aculeatus, Cronobacter malonaticus, C. sakazakii, Klebsiella pneumoniae, Acetobacter sp., Serratia marcescens, Enterobacter hormaechei, Staphylococcus saprophyticus, Lactococcus garvieae, Lactobacillus plantarum, Lysinibacillus fusiformis, Lysinibacillus sp., Bacillus amyloliquefaciens, B. cereus, Bacillus sp.- 1, Bacillus sp.- 2 using the separated method.

The incidence and McKinney index of 15 bacterial species and 10 fungal species were significantly different from those of the control (sterile water and LB medium) using the merged placement method (incidence: F = 10.44, P < 0.01; McKinney index: F = 43.28, P < 0.01; Fig. 5A and Fig. 5B). Fungal isolates demonstrated stronger spoilage potential in the grape berries with an incidence of more than 75%. Except for C. oxysporum and P. citrinum, the McKinney index of all other fungi exceeded 50%, which was greater than that of bacteria. Three Aspergillus species and P. georgiense showed 100% spoilage on the McKinney index. Healthy grapes were also highly sensitive to A. tenuissima and F. proliferatum, which high McKinney index values of 52 ± 1 and 50 ± 2, respectively. Among the bacteria, the incidence and McKinney index of two Cronobacter species, Serratia marcescens and Lysinibacillus fusiformis, were higher than those of the other bacteria. B. amyloliquefaciens and B. cereus led to less serious spoilage than other bacteria (Fig. 4, then Fig. 5). The percentages of incidence and McKinney index of microorganism were lower using separated infection method than merged method, while those of fungi were also higher than bacteria (Percentage of incidence: F = 90.52, P < 0.01; McKinney index: F = 50.49, P < 0.01; Fig. 5C and Fig. 5D). The C. oxysporum had the highest spoilage potential among microbial taxa, which was different from the results obtained under merged infection.

Figure 5 The percentage of incidence of microorganism using merged (A) or separated (C) methods; McKinney index of microorganism using merged (B) or separated (D) methods.

Different letters in each figure indicate significant difference between microorganisms (one-way ANOVA; α = 0.05).

Discussion

Metagenomic analysis and culturing indicated that the microorganisms caused spoilage were similar around the world. AAB were the dominant bacteria in rot-affected grapes in eastern coastal areas of China, which is consistent with reports from Australia (Mateo et al., 2014), Portugal (Barata, Malfeito-Ferreira & Loureiro, 2012b) and New York (Hall, O’Bryon & Osier, 2019). Aspergillis were the dominant mold, which agrees with reports from Greece (Tjamos et al., 2004) and California (Rooney-Latham et al., 2008). Issatchenkia occidentalis, Hanseniaspora uvarum, and Candida vanderwaltii, Colletotrichum viniferum, and Saprochaete gigas, were also commonly observed, which are the same genera, but different species, than reports from other regions (Guerzoni & Marchetti, 1987; Barata et al., 2008; Barata, Malfeito-Ferreira & Loureiro, 2012b; Lleixa et al., 2018). Pisani, Nguyen & Gubler (2015) reported that grape sour rot is a disease complex involving many filamentous fungi and bacteria but is usually initiated by A. niger or A. carbonarius in California. We observed similar communities and reported the presence of pathogens, that could infect humans and animals, associated with rotting grapes, such as C. sakazakii SRG2, K. pneumoniae SRG3, and S. gigas SRG18. C. sakazakii is an emerging opportunistic foodborne pathogen with the potential to cause meningitis, bacteremia, and necrotizing enterocolitis, particularly in infants (Drudy et al., 2006; Aly et al., 2019). K. pneumoniae is an important conditional pathogenic and iatrogenic infectious bacterium. Saprochaete yeasts have emerged as fungal pathogens and causal agents of life-threatening infections in patients with severe neutropenia and hematological malignancies (Pavone et al., 2019). Therefore, sour rotten berries could be a reservoir for human pathogens.

Fungal isolates demonstrated greater spoilage potential than bacterial isolates in the grape berries. Except for three Aspergillus species with high McKinney index values, healthy grapes were also sensitive to spoilage fungi (A. tenuissima and F. proliferatum) associated with common grape diseases, which was different from studies in other places. As the most common species in the cosmopolitan genus Alternaria, A. tenuissima is found on a broad range of fruit products and causes various diseases, like post-harvest black rot of fruit (Logrieco, Moretti & Solfrizzo, 2009). Bakshi, Sztejnberg & Yarden (2001) reported that F. proliferatum could also cause the rot of corn, rice, and lily. Therefore, Aspergillus species, A. tenuissima, and F. proliferatum were the main cultivated spoilage fungi causing sour rot in grapes. Among the bacterial isolates, B. amyloliquefaciens and B. cereus led to less serious sour rot in this study. This can possibly be explained by the antibacterial substances generated by B. amyloliquefaciens and B. cereus, which have been used as biological control agents (Risoen, Ronning & Hegna, 2004; Wang et al., 2014). Although L. fusiformis restricts the biofilm formation of some pathogenic bacteria, it caused serious rot in grape berries (Fig. 3). Healthy grapes were sensitive to Cronobacter sp. and S. marcescens, which are spoilage microorganisms (Healy et al., 2010). The spoilage potential assay confirmed that Cronobacter species, S. marcescens, and L. fusiformis can cause sour rot in grapes. In this study, the incidence and McKinney index of microorganisms were lower using the separate infection method than using the merged method, further suggesting that diseases related to sour rotten grapes could spread quickly through grape clusters.

Sour rot is the culmination of coinfection by various yeasts that convert grape sugars to ethanol and bacteria that oxidize the ethanol to acetic acid (Pinto et al., 2019), and Drosophila spp. mediate these processes (Hall et al., 2018). Sour rot increases attractiveness to ovipositing D. melanogaster females and oviposition by D. suzukii facilitates sour rot development (Rombaut et al., 2017; Ioriatti et al., 2018). Furthermore, musts and the beginning of fermentation using rotten Macabeo grapes is consistently characterized by an elevated frequency of Zygosaccharomyces, and AAB increase in the late stages of fermentation (Lleixa et al., 2018). It is difficult to control sour rot in grapes due to the multiple species associated with this disease. Therefore, relationships among insects, microorganisms, and grapes as well as comprehensive analyses of nosogenesis will be the key question in the researches of sour rot in grapes.

Conclusions

This study identified more spoilage species in sour rot-affected grapes of China using culture-dependent methods combined with high-throughput sequencing analysis, which would provide comprehensive information on targets for the control of the disease. Majority of these microbes could infect grapes with wounds. The microbes associated with sour grape rot in eastern coastal China appear similar to those associated with this disease in vineyards around the world. We reported here that A. tenuissima, and F. proliferatum spoil grapes. Human and animal pathogens were also present among the bacteria in sour rot-affected grapes, such as Cronobacter sakazakii, Klebsiella pneumoniae and S. gigas.

Supplemental Information

Table S1 Raw data for spoilage potential of microorganism using two methods

The McKinney index and percentage of incidence for spoilage potential of microorganism with three replicates using merged and separated methods

Click here for additional data file.

Figure S1 The rarefaction curves and abundance-OUT rank curves of 16S rRNA and ITS sequences based on 16S rDNA high-throughput sequencing

The rarefaction curvesand abundance-OUT rank curves of 16S rRNA and ITS sequences based on 16S rDNA high-throughput sequencing. The curves of three samples tended to be flat in rarefaction curves, which indicated that the amount of sequential data of three samples were reasonable. The flat curves of abundance-OUT rank indicated a high degree of sequencing uniformity

Click here for additional data file.

Figure S2 The first 50 OTUs of the bacteria by high-throughput sequencing

The first 50 OTUs of the bacteria by high-throughput sequencing

Click here for additional data file.

Figure S3 The first 50 OTUs of the fungi by high-throughput sequencing

Click here for additional data file.

Data S1 The raw data of 16S rDNA through high throughput sequencing from 5’ end

The original image data file of 16S rDNA obtained by Illumina Miseq™ from 5’ end was transformed into the original sequencing sequence by CASAVA Base Calling analysis (Sequenced Reads).

Click here for additional data file.

Data S2 The raw data of 16S rDNA through high throughput sequencing from 3’ end

The original image data file of 16S rDNA obtained by Illumina Miseq™ from 3’ end was transformed into the original sequencing sequence by CASAVA Base Calling analysis (Sequenced Reads).

Click here for additional data file.

Data S3 The raw data of ITS through high throughput sequencing

The original image data file of ITS gene obtained by Illumina Miseq™ was transformed into the original sequencing sequence by CASAVA Base Calling analysis (Sequenced Reads).

Click here for additional data file.

Data S4 Sequences of bacteria in sour rotten grape

Click here for additional data file.

Data S5 Sequences of fungi in sour rotten grape

Click here for additional data file.

We would like to thank Chao Li for assistance with loading sequences of fungi, Dongyun Qin and Ling Su for assistance with grape collection, and Sha Liu and Dongyun Qin for assistance with the treatment of grape samples.

Additional Information and Declarations

Competing Interests

Author Contributions

DNA Deposition

Data Availability

The authors declare there are no competing interests.

Huanhuan Gao and Xiangtian Yin conceived and designed the experiments, performed the experiments, analyzed the data, prepared figures and/or tables, and approved the final draft.

Xilong Jiang performed the experiments, analyzed the data, authored or reviewed drafts of the paper, and approved the final draft.

Hongmei Shi, Chaoping Wang and Yingchun Chen performed the experiments, prepared figures and/or tables, and approved the final draft.

Yang Yang analyzed the data, prepared figures and/or tables, and approved the final draft.

Xiaoyan Dai performed the experiments, authored or reviewed drafts of the paper, and approved the final draft.

Xinying Wu analyzed the data, authored or reviewed drafts of the paper, and approved the final draft.

The following information was supplied regarding the deposition of DNA sequences:

The group sequences of bacteria are available at GenBank: MK743980 to MK743994 and MK753026. The group sequences of fungi are available at GenBank: MK748309 to MK748317. The Saprochete gigas sequence is available at GenBank: MN567950.

The following information was supplied regarding data availability:

The raw measurements are available in the Supplementary Files.

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
