# Peer review of "Diversity and spoilage potential of microbial communities associated with grape sour rot in eastern coastal areas of China"

_PeerJ, doi:10.7717/peerj.9376_

## Round 0.1 · original submission · Major Revisions

Diversity and pathogenicity of microbial communities causing grape sour rot in eastern coastal areas of China

The reviewers all expressed an interest in the work. One reviewer recommended rejection but I think it is possible to address their concerns if the paper is revised following the three reviewer’s suggestions. I could make many edits to the text but at this time I provide examples from the abstract to guide the revision process:

Line 26. Revise to “..sour rot decreases the yield..”
Line 29-30. Background in Abstract should state the goal of the study. Also, delete “and even in the world.”
Line 32. Delete “pathogenic.” The focus of work appears to be identifying microbes associated with but not necessarily causing rot.
Line 34. Revise to assessed by “culture-dependent and metagenomic analysis of…”
Line 35. Delete “then”

Reviewer 1 ·

Basic reporting

1 BASIC REPORTING
Clear, unambiguous, professional English language used throughout.
The English language used throughout the manuscript is clear and unambiguous. Some words used in the title and throughout the manuscript are not technically correct. For example, “pathogenicity” and “pathogenic” should be replaced by “spoilage potential” and “spoilage”, respectively. Moreover, the terms “morbidity” and “disease index” should be replaced by “percentage of incidence” and “McKinney index”. Finally, the terms “cultivable” and “cultivated bacteria and fungi” could be replaced by “culturable” and “culturable microbial community”.
Intro & background to show context. Literature well referenced & relevant.
The introduction needs revision. The section at lines 86-92 could be shortened. I suggest to remove lines 93-101 (in my opinion the literature cited is not relevant for this work) and to replace this part with metagenomics studies or studies using culture-independent approaches related to grape rots (in particular sour rot) and wine and table grape microbiota.
Structure conforms to PeerJ Standards, discipline norm, or improved for clarity.
The structure conforms to PeerJ Standards.
Figures are relevant, high quality, well labelled & described.
Figures are relevant, of good quality, well labelled and described as shown by figure captions and in the main text.
Raw data supplied.
Raw data are supplied in the supplementary files.

Experimental design

2 EXPERIMENTAL DESIGN
Original primary research within Scope of the journal.
Research is in accordance with Scope of the journal.
Research question well defined, relevant & meaningful. It is stated how the research fills and identified knowledge gap.
Research objectives are well defined, relevant, and meaningful:
The objective of this study was to determine: (1) the diversity and abundance of bacteria and fungi in sour rot-infected table grapes collected from Yantai city based on traditional culture methods; (2) the pathogenicity of bacteria and fungi associated with grape sour rot; and (3) the diversity and abundance of bacteria and fungi in sour rot-infected table grapes based on 16S rRNA and ITS high-throughput sequencing analysis.
In my opinion the workflow of the work could be revised. In particular, the spoilage potential of bacteria and fungi associated with grape sour rot could be presented as the third step of the work, while the high-throughput sequencing analysis could follow the characterization of culturable microorganisms. This order should be followed throughout the manuscript.
As regards statement related to the fill of research gap, this part is reported at lines 81-85. I suggest to reinforce this part and to report the statement just before the objectives at the end of Introduction section.
Rigorous investigation performed to a high technical & ethical standard.
Research activities are performed following high technical and ethical standard.
Methods described with sufficient detail & information to replicate.
The methods are described with sufficient information regarding procedures and materials used.

Validity of the findings

3 VALIDITY OF THE FINDINGS
Impact and novelty not assessed. Negative/inconclusive results accepted. Meaningful replication encouraged where rationale & benefit to literature is clearly stated.
Decisions are made based on impact and novelty of interest to a wide audience. Nowadays, the microbiota of rotten grape berries is limited characterized, so studies related to this field are appreciated and encouraged by the scientific community.
As regards replication, the authors analysed the data from three technical replicates for different analyses. In this kind of study biological replicates should be considered. Grapes of the same variety from different vineyards or vines, and collected in different vintages should be analysed. However, I understand that this kind of sampling is difficult and time-consuming. I think that the goal of this work is the microbial characterization of the table grape sour rot microbiota and mycobiota using two different approaches and not the study of these microorganisms from different cultivation sites and vintages.
All underlying data have been provided; they are robust, statistically sound & controlled.
Data have been provided and they are relevant, statistically analysed and checked.
Conclusions are well stated, linked to original research question & limited to supporting results.
Conclusions are well stated and linked to original research questions. However, I suggest to introduce other findings such the isolation from rotten berries of bacteria such as Cronobacter sakazakii, an emergent pathogen, and Klebsiella pneumoniae, a source and shuttle for antibiotic resistance.
The conclusions as well as the entire manuscript should report only supported results. Sour rot symptoms are well defined: skin browning, ethyl acetate smell, disaggregation of the internal tissues, berry emptying. Sour rot symptoms can be reproduced at lab level using Drosophila flies (doi.org/10.1007/s00248-012-0041-y), or by selected NSY-AAB associations (doi.org/10.1016/j.fm.2017.05.001) that are naturally carried out on grapes by the insects.
The symptoms in figure 3 are not related to sour rot but are examples of spoilage caused by sour rot bacteria and fungi on berries (cracking, colonization of grape tissue by bacteria, or mycelium development on the grape berry surface). Acetobacter sp. did not develop sour rot symptoms on berries probably because the sour development needs the presence of Drosophila flies to prevent wound healing or the co-occurrence of selected yeasts. The etiology of sour rot is still determined by acetic acid bacteria and non-Saccharomyces yeasts, or by Aspergillus spp. that the authors detected by high-throughput sequencing.
However, the authors demonstrated that different bacteria and fungi isolated from sour rotten berries could be involved in grape spoilage. It is an important finding that could be reported in the conclusions.
I suggest to revise the Conclusions section.
Speculation is welcome, but should be identified as such.
See reviewer comments reported above.

Additional comments

4 GENERAL COMMENTS
The work entitled “Diversity and pathogenicity of microbial communities causing grape sour rot in eastern coastal areas of China” reports the genetic and physiological characterization of culturable fungi and bacteria from sour rotten berries, the description of microbial communities of rotten berries by high-throughput sequencing, and the spoilage potential of isolated fungi and bacteria on grape berries. The study is relevant for a broad scientific community since it describe for the first time the isolation of certain bacteria and fungi from grape sour rot. In addition, the spoilage potential of the isolated microorganisms is presented. However, the weakness of the work is the casual relationship between the isolation of certain microorganisms and the development of the disease. Moreover, a revision of the introduction and discussion section is recommended. English language is of sufficient quality but a revision of technical words is suggested.

Specific comments
Title: Diversity and pathogenicity of microbial communities causing grape sour rot in eastern coastal areas of China
The title did not reflect the results. Microbial communities characterized from sour rotten berries could not be involved in the development of sour rot. I suggest to change the title as follow:
Diversity and spoilage potential of microbial communities of grape sour rot in eastern coastal areas of China
Abstract
L28 “Planting location” and “identified methods” should be replaced, here and throughout the manuscript, by cultivation site and microbial identification methods.
Please, as suggested in the basic reporting, not use the terms pathogens, pathogenicity, pathogenic. It is better spoilage microorganisms, spoilage potential.
L34-L38 Here and throughout the manuscript please replace culture-methods with culture dependent methods.
L48 Incertae sedis sp. is an unknown or undefined taxonomic classification. Thus, it could be removed from the list of fungi and yeasts identified in rotten berries.
Introduction
Introduction section should be revised as requested in the basic reporting part.
L68, L342, L444 Please correct the surname Guerzpni. The correct surname is Guerzoni.
L95 18S rRNA? I did not find 18S rRNA sequencing in Materials and Methods. Are you sure?
Materials and Methods
L113-119 Which grape variety was used?
Which is the total weight of grape tissue collected? 1g or 100 g from 100 rotten grapes? How did you perform the collection?
L121 The samples … Which is the weight of each replicate? Which is the dilution ratio?
L123-124 Did the authors use antibiotics in the medium to prevent the cross-contamination (bacteria on PDA and fungi on nutrient agar)?
L126 ….using the primary medium
L133 …DNA from single colony of each bacterium..
L137 Is the term “quantification” correct?
L153-156 The authors cloned the PCR products. Did the authors find multiple bands on gel? Was this approach used to be sure to work with a single copy of target gene?
L169 …106 conidia/mL or cfu/mL in the case of bacteria.
L202 Remove “that were” and add “and”.
Results
L249-250 It’s better “Sequences of these 15 bacterial species and 10 fungal species showed a 97-100% of similarity with those deposited in NCBI”.
L258 Please replace diseased with spoiled
L260-272 Please use the terms percentage of incidence and McKinney index as suggested in Basic Reporting.
L288-289 I see the Table 3. The microbial diversity (Shannon index and Simpson index) is higher for bacteria than fungi. The richness (Chao1 index and ACE index) is higher for fungi than bacteria. Is it correct? Please verify.
.L298-301 I suggest to describe the dominant species and then that 32.40% of the phyla were not taxonomically classified (Incertae sedis sp.).
L302-313 Which is the prediction accuracy based on yours OTU number/sample of PICRUSt predictions? Which is the accuracy for each functional category in COG and KEGG classification?
Discussion
The Discussion section needs integration. In particular three parts should be added:
1) metagenomic studies on sour rot bacterial and fungal biota.
2) description of new species isolated from rotten berries. For example Cronobacter species, Klebsiella pneumoniae, and Saprochaete gigas. Cronobacter species are emergent pathogens; K. pneumoniae is a multidrug resistant pathogen; S. gigas is of the same genera of Saprochaete capitata, an emerging fungal pathogen with low susceptibility to echinocandins. Therefore, sour rotten berries could be a reservouir of human pathogens and antibiotic resistance potentially transferable to other bacteria.
3) a discussion on functional classification based on COG and KEGG analysis. In particular, the transport of metabolites, coenzymes, the membrane transport, carbohydrate and amino acid metabolisms are the principal functions. Indeed, these functions are typical of AAB, the main bacterial group found by high-throughput sequencing. The authors could briefly discuss the AAB metabolisms in sour rotten berries (oxidative fermentation, production of acetic acid, amino acid metabolism and so on).

Reviewer 2 ·

Basic reporting

This manuscript reports on the characterisation of the microbiota related to the polymicrobial disease grape sour rot.
To identify the causative agents of the disease the authors present a set of microbiological tool and genetic tools. They have studied the non-cultivable microbiome applying next generation sequencing through Illumina MiSeq. In addition they have isolated the bacterial and fungal cultivable microbiome and identified the causative agents by two independent methods, characterising the bacterial phenotype and the taxonomical features of the fungi, and ultimately bacteria and fungi isolated were also genetically characterised by sequencing of plasmids by Sanger sequencing of 16S rRNA gene and internal transcribed spacer (ITS) respectively.

Experimental design

The relevance to disease is well known therefore the research poses interest for the scientific community, however there is a number of issues that make me recommend the rejection of this paper.

1. It is too risky to identify to the level of species bacteria or fungi through metagenomics. Even Sanger sequencing is not an accurate method when just based in the sequencing of one gene. For fungi for instance the authors should consider to include additional genomic markers such us RNA polymerase subunit II (RPB2) (Thongklang et al., 2014) or the partial large sub- unit (LSU) of ribosomal DNA (Geml et al., 2004), ITS is a highly variable region.
2. The number of causative agents presented is significantly high and not effectively discussed to consider that pathogenicity tests are efficient to consider all the parasites reported as causative agent of grape sour rot or, in the other hand, some other disease. For instance, the authors cite Fusarium proliferatum as parasite, however, they never discuss that this same species has been reported as an effective biocontrol agent to fight grape downy mildew, Plasmopara viticola (Falk et al., 1996; Baksi et al., 2001), which is remarkable to me. The trial desing of pathogenic tests as reflected in Figure 3 can induce cross contamination due to the proximity of the fruits.
3. The methodology employed for the construction of the libraries sequenced in MiSeq is non decribed. How many replicates do they sequence? What kind of sample are they sequencing, how did they extract DNA, PCR amplification…? Which is the final concentration of the pool? What kind of control are they introducing in the pool? What is the kit used? What kind of indexing did they performed? How did they check the quality of the library before running?All these technical issues required to reproduced the trial are missing.
4. Results of NGS sequencing. Non data concerning quality of the sequencing is presented. Quality achieved is missing, number of reads per sample is missing. The mean size of the 16S rRNA (412 bp, is too low, should be around 550 bp before indexing, 620 bp afterwards) fragments and ITS (450 bp, is too low, should be around 450 bp before indexing, 560 bp afterwards) are not accurate to me.
5. Discussion is poor and difficult to follow. While presenting a good amount of work, they just attached 33 references, what makes me think that a poor approach to the state-of-the-art was performed before and after the trials. For instance I found a lot of literature related to grape sour rot and however the authors sometimes reflect other crops, such as in lines 352-356 when they talk about parasites related to fruits, corn, rice or lily, this is out of scope and not helpful for their work.
6. Figures and tables. There are so many figures and table, sometimes less is more, I have to ask the authors if all these 9 figures they present are mandatory to expose their results. I think they need to think about this issue. Maybe supplementary material is a good place to fit some of them.
7. References. As cited above, literature cited is poor, besides 12 out of 33 references are very old, over 10 years, I think we need to make an effort to update the science continuously, and try to be focused within the last ten years (no more than 10% of cites older than that).

Validity of the findings

Due to the number of issues cited and some others I recommend the editor to reject this paper and encourage the authors to fully refurbish their work and try to submit it to the suitable journal.

Additional comments

This manuscript reports on the characterisation of the microbiota related to the polymicrobial disease grape sour rot.
To identify the causative agents of the disease the authors present a set of microbiological tool and genetic tools. They have studied the non-cultivable microbiome applying next generation sequencing through Illumina MiSeq. In addition they have isolated the bacterial and fungal cultivable microbiome and identified the causative agents by two independent methods, characterising the bacterial phenotype and the taxonomical features of the fungi, and ultimately bacteria and fungi isolated were also genetically characterised by sequencing of plasmids by Sanger sequencing of 16S rRNA gene and internal transcribed spacer (ITS) respectively.
The relevance to disease is well known therefore the research poses interest for the scientific community, however there is a number of issues that make me recommend the rejection of this paper.

1. It is too risky to identify to the level of species bacteria or fungi through metagenomics. Even Sanger sequencing is not an accurate method when just based in the sequencing of one gene. For fungi for instance the authors should consider to include additional genomic markers such us RNA polymerase subunit II (RPB2) (Thongklang et al., 2014) or the partial large sub- unit (LSU) of ribosomal DNA (Geml et al., 2004), ITS is a highly variable region.
2. The number of causative agents presented is significantly high and not effectively discussed to consider that pathogenicity tests are efficient to consider all the parasites reported as causative agent of grape sour rot or, in the other hand, some other disease. For instance, the authors cite Fusarium proliferatum as parasite, however, they never discuss that this same species has been reported as an effective biocontrol agent to fight grape downy mildew, Plasmopara viticola (Falk et al., 1996; Baksi et al., 2001), which is remarkable to me. The trial desing of pathogenic tests as reflected in Figure 3 can induce cross contamination due to the proximity of the fruits.
3. The methodology employed for the construction of the libraries sequenced in MiSeq is non decribed. How many replicates do they sequence? What kind of sample are they sequencing, how did they extract DNA, PCR amplification…? Which is the final concentration of the pool? What kind of control are they introducing in the pool? What is the kit used? What kind of indexing did they performed? How did they check the quality of the library before running?All these technical issues required to reproduced the trial are missing.
4. Results of NGS sequencing. Non data concerning quality of the sequencing is presented. Quality achieved is missing, number of reads per sample is missing. The mean size of the 16S rRNA (412 bp, is too low, should be around 550 bp before indexing, 620 bp afterwards) fragments and ITS (450 bp, is too low, should be around 450 bp before indexing, 560 bp afterwards) are not accurate to me.
5. Discussion is poor and difficult to follow. While presenting a good amount of work, they just attached 33 references, what makes me think that a poor approach to the state-of-the-art was performed before and after the trials. For instance I found a lot of literature related to grape sour rot and however the authors sometimes reflect other crops, such as in lines 352-356 when they talk about parasites related to fruits, corn, rice or lily, this is out of scope and not helpful for their work.
6. Figures and tables. There are so many figures and table, sometimes less is more, I have to ask the authors if all these 9 figures they present are mandatory to expose their results. I think they need to think about this issue. Maybe supplementary material is a good place to fit some of them.
7. References. As cited above, literature cited is poor, besides 12 out of 33 references are very old, over 10 years, I think we need to make an effort to update the science continuously, and try to be focused within the last ten years (no more than 10% of cites older than that).

Due to the number of issues cited and some others I recommend the editor to reject this paper and encourage the authors to fully refurbish their work and try to submit it to the suitable journal.

·

Basic reporting

Research into sour rot-associated microorganisms is very relevant, but this article is not updated with the current literature. Four papers in the past year have been published specifically into sour rot causal organisms and associated microorganisms, none of which are mentioned in this paper, and which would significantly impact the interpretation of the results.

Experimental design

The assessment of sour rot infection on inoculated berries is questionable, because the authors seem only be assessing the berries based on area of the berry that is infected. This is not an appropriate assessment for sour rot because it does not take into account the loss of berry integrity that is a pivotal sign of the disease. Then, to assess causal organisms based on an incomplete rating system does not actually determine causality, but instead, association. The authors of determined associated microorganisms, not casual microorganisms. Because it is mentioned that the culture methods are less successful at identifying microorganisms than high-throughput sequencing methods, why is it necessary to include the culture methods?

Validity of the findings

The validity of the findings is questionable because the authors describe these organisms as causal, not as associated. There is little evidence that they are causal organisms of this disease when the method of assessing disease symptoms is incomplete. It would be beneficial to rephrase the purpose of the article as looking at associated microorganisms instead of causal microorganisms.

---

## Round 0.2 · Minor Revisions

If the reviewer's comments are adequately addressed, I do not think we will need to send the manuscript out for review.

Reviewer 1 ·

Basic reporting

1 BASIC REPORTING
Clear, unambiguous, professional English language used throughout.
The English language used throughout the manuscript is clear and unambiguous.
Intro & background to show context. Literature well referenced & relevant.
The introduction section has been improved. In the references Pinto et al. (2019) is lacking. Please verify
Structure conforms to PeerJ Standards, discipline norm, or improved for clarity.
The structure conforms to PeerJ Standards.
Figures are relevant, high quality, well labelled & described.
Figures are relevant, of good quality, well labelled and described as shown by figure captions and in the main text.
Raw data supplied.
Raw data are supplied in the supplementary files.

Experimental design

Original primary research within Scope of the journal.
Research is in accordance with Scope of the journal.
Research question well defined, relevant & meaningful. It is stated how the research fills and identified knowledge gap.
Research objectives are well defined, relevant, and meaningful:
The workflow has been revised.
Rigorous investigation performed to a high technical & ethical standard.
Research activities are performed following high technical and ethical standard.
Methods described with sufficient detail & information to replicate.
The methods are described with sufficient information regarding procedures and materials used. However, I suggest to add more information on functional analysis.

Validity of the findings

Impact and novelty not assessed. Negative/inconclusive results accepted. Meaningful replication encouraged where rationale & benefit to literature is clearly stated.
All underlying data have been provided; they are robust, statistically sound & controlled.
Data have been provided and they are relevant, statistically analysed and checked.
Conclusions are well stated, linked to original research question & limited to supporting results.
Conclusions have been revised.
Speculation is welcome, but should be identified as such.
No comment

Additional comments

The manuscript has been improved as suggested by the reviewer. It is acceptable for publication in “Peer J” after minor revisions.
Specific comments
1
Please, throughout the manuscript, check spaces, italic typesetting, capital and lowercase letters, removed equations, font style and so on: L73-74, L84, L83-85, L98, L111, L144-L145, L208, L230-1, L234, L248-9, L255-60, L288, L292, L299, L311-12, L330, L353, L360, L396-99, L408-9, Table 3, Table 4. Please add Pinto et al. (2019) to References.
2
The symptoms in figure 3 are not related to sour rot but are examples of spoilage caused by sour rot bacteria and fungi on berries (cracking, colonization of grape tissue by bacteria, or mycelium development on the grape berry surface). Acetobacter sp. did not develop sour rot symptoms on berries probably because the sour development needs the presence of Drosophila flies to prevent wound healing or the co-occurrence of selected yeasts. The etiology of sour rot is still determined by acetic acid bacteria and non-Saccharomyces yeasts, or by Aspergillus spp. that the authors detected by high-throughput sequencing.
Respond: Thanks for your comments, in this experiment, we simulated cracked fruit and feeding of fruit flies and made the wound using needle. This proves that Acetobacter sp. can lead to the symptoms of sour rotted grape in the presence of wounds.

Reviewer comment:

Dear authors,
as I can see in the figures 5 A-16 and 5 B-16 Acetobacter sp. did not develop sour rot symptoms on table grape berries. The figures show spoilage symptoms caused by bacterial colonization but they are far from the typical sour rot. I understand that the reproduction of sour rot symptoms under laboratory conditions is difficult, especially for red skin table grapes. I suggest (for future studies) to better characterize Acetobacter sp. strain through ITS sequencing (in this work the taxonomic position is questionable), sour rot development on grape berries (I suggest to inoculate the strain -10 µL at 108 cfu/mL- in the pedicel cavity after pedicel remove instead in wounds made with needle) and biochemical characteristics.
3
L302-313 Which is the prediction accuracy based on yours OTU number/sample of PICRUSt predictions? Which is the accuracy for each functional category in COG and KEGG classification?

Respond: In our study, three samples were analyzed and 16S rRNA in the samples was annotated according to the COG and KEGG data base. However, the figure and result was the merged result of three samples (replicates). Therefore, no variation analysis was showed in the manuscript. Could you please tell me what is the prediction accuracy? Which result needs to be supplemented, thank you very much.

Reviewer comment:

Dear authors,
16S rRNA in the samples was annotated according to the COG and KEGG data base, using the software PICRUSt. This software works with predictions that can be not accurate. I strongly suggest reading the following document:
https://picrust.github.io/picrust/tutorials/quality_control.html
For these reasons, authors should provide more information related to functional analysis in Materials and Methods and Results. In particular:

- For the standard metagenome prediction workflow, PICRUSt requires that input 16S rRNA data be in the form of a BIOM format table that was picked by mapping your reads to references in the greengenes tree. The most straightforward way to do this is using QIIME’s reference-based OTU picking pipeline as described in the OTU Picking Tutorial. Therefore, OTU IDs must be based on greengenes and sequences that fail to map to references will not be predicted. If the authors used this workflow please add in Materials and Methods.

- Are reference genomes available for the more abundant bacterial species? Please add this information in Results

- Calculate reference genome coverage for your samples using NSTI scores. Please add in Results

- Calculate metagenomics confidence intervals. Please add in Results

- Some details of the reference-based OTU-picking step may be useful to report in order to ensure reproducibility of the results. These include: the version of the Greengenes reference used, the proportion of reads that mapped to reference during OTU picking, the similarity threshold used during OTU-picking. Please include these details in Materials and Methods.

·

Basic reporting

No comment.

Experimental design

No comment

Validity of the findings

No comment

Additional comments

Lines 60-63: Provide citation.
Lines 63-64: Provide citation or remove.
All references to sour rot-infected should be changed to sour rot-affected. Sour rot is not a pathogen, it is a disease, and can therefore not infect a grape, but it can affect one.

---

## Round 0.3 · Minor Revisions

Before I can send the manuscript for review, please address the following issues in Abstract and make the corresponding changes in the body of the manuscript, which I will review after Abstract is revised. That is the comments below should apply to the entire manuscript.

Line 27. Background of Abstract still lacks a justification. I understand they grow a lot of grapes in China, and rot destroys a lot of the crop, but what is not known?
Line 32. Revise to “was determined with an in vitro spoilage assay.”
Line 34. Write more succinctly. Revise to “We isolated 15 bacterial species and…”
Line 39. I do not know what “firstly discovered as spoilage microorganisms..” means. Are you saying this is the first evidence that these species spoil grapes? Move that statement to conclusions and revise for clarity. Also, delete “Moreover,” which adds nothing.
Line 40. OTUs counts must be whole numbers. 1343.33 species makes not sense. Also, the word “revealed” doesn’t work here.
Line 41. Present reasonable significant figures (72% not 72.15%).
Line 45. The word “took” doesn’t work here.
Line 46-48. This is a weak conclusion. Are you saying more accurate results could have been obtained if you had used better methods. Conclusion, should relate to the goal of the project. Did you identify the microbes responsible for rotting grapes in China?
Also, carefully proofread references. As submitted, they do not have a consistent format. For example, only cap proper nouns in manuscript titles (line 411, 452), do not include the issue (line 431) and italicize genus and species (line 435, 444).

---

## Round 0.4 · Major Revisions

The manuscript still needs substantial review for style, as my comments have been only partially addressed.

Line 23. Italicize “in vitro”
Line 35. Delete “Based on …methods”
Line 39. Replace “) as mainly …grape. The results of high-” with “) spoiled grapes. High-“
Line 40. Delete “the OTUs….respectively”
Line 47. Replace “grapes. It was… A. termussima” with “grapes and A.termussima”
Line 53. Delete “often”
Line 56. Replace “are often considered as the causes” with “can cause”
Line 59. Replace “Studies…For example, the most..” with “The most..”
Line 66. Avoid passive voice revise to “Basidomycetes.. dominate intact grape”
Line 72. Present established science in present tense. Replace “could also be identified” with “occur”
Line 75 -77. Delete “Although… Brady et al. 2017).”
Line 83. Revise to “…sequencing and found”
Line 84-85. Delete “yeast genera..in both sets of samples”
Line 88-91. Replace “was used in the present…this study was to determine” with “was used to determine”
Line 93. Replace with “spoilage potential of these microbes with sour rot of grapes”
Line 102. How many 50 ml conical tubes? One or three?
Line 172. Provide reference for Mothur.
Line 200. Use integers for area infected (50% not 50.1%).
Line 217. Revise to “We isolated six species of Proteobacteria..”
Line 218. Delete “were also..Table 2.”
Line 223. Replace “Among…fungi” with “We cultured ten fungi species”
Line 225. Delete “The characteristics…in Fig. 4.”
Line 237. Revise to “these isolates showed >97 % similarity..”
Line 242-244. Delete “We performed…However.”
Line 245. Present read lengths in integers (5 not 4.52) and delete “which was shorter than that fo”
Line 247. Revise to “…(Table 3). Rarefaction curves….reached asymptopes…”
Line 252. Delete “Phylogenetic trees..of sequences.”
Line 259. Delete “The diversity …Table 4.”
Line 263. Delete “The bacterial..Fig. 3”
Line 266. Use integers here (38% not 37.62%)
Line 268. Delete “The fungal..is shown in”
Line 275. Delete “Each bacterial…Midnight Beauty”
Line 305. Replace “ (AAB) are usually…causing” with “(AAB) cause”
Line 310. Delete “through molecular…methods”
Line 312. Revise to “14 other species were isolated from sour grapes”
Line 315. Use integers here.
Line 336-340. Delete “Although…to echinocandins.”
Line 340. Revise to “have emerged as fungal”
Line 347. Use integers here.
Line 350-352. Delete “bacteria and fungi…grapes.”
Line 367. Revise to “L. fusiformis can cause sour…”
Line 373. Use active voice. Revise to “Drosophila mediates this process”
Line 380. Delete “However..of sour rot.”
Line 392-402. Delete entire Conclusions.
Table 3. Present reads as integers (447 not 446.72) and delete means.
Table 4. Replace “OTUs…number of” with “Diversity indices of” and present ACE, Chao1 and OTUs as integers.
Figure 1 legend. Replace “lables” with “labels”
Fig 3 legend. Insert space in “ongenus”
Fig 4 insert space in “onphylum”

Reviewer 1 ·

Basic reporting

No comment

Experimental design

No comment

Validity of the findings

No comment

Additional comments

Dear authors,
the manuscript is acceptable for publication in Peer J.

---

## Round 0.5 · Minor Revisions

The manuscript still needs substantial review for style. I provide some examples below but this manuscript would benefit from the professional editing service.

Line 26. Delete “and the ..methods.” Methods don’t determine diversity. They measure it.
Line 27. Replace “,,in China …has not reported.” With “in China; the microbes responsible for causing sour rot of grapes in this area have not been identified.”
Line 32. Replace with “..ITS genes with high-“
Line 33. Delete “spoilage”
Line 36. Replace with “confirmed that five isolated fungal strains … and four isolated bacterial strains.. could spoil grapes.”
Line 39. Replace with “…sequencing identified 1343 OTUs of bacteria and 1038 OTUs of fungi.”
Line 41. Delete “while” and “Then”
Line 42. Avoid phrase “such as”
Line 44. Replace “in sour rot-“ with “associated with sour rot-“
Line 44-47. Replace “In conclusion…harvesting grape.” With “This suggests that multiple bacterial and fungal species can cause sour rot of grapes, including A. tenuissima and F. proliferatum.”
Line 57-58. Replace “Studies have analyzed…grape cultivars.” With “The frequency and density of yeast species associated with sour rot differs between grape cultivars. The most frequently recovered ascomycetes species are…”
Line 62. Here and throughout, avoid starting sentences with “moreover” or “however,” or phrases like “in view of this,” which add no content.
Line 66. Define AAB and revise to “AAB dominate rotten grapes.”
Line 74. Delete “excluded the limitations…samples.”
Line 76. Delete “about the research”
Line 204. Use the same sentence structure for this topic sentence as the next paragraph. Revise to “We cultured 15 bacterial strains from…”
Line 208. How can Proteobacteria be Gram positive? Also, Gram is a proper noun.
Line 225-227. You cannot have a paragraph that simply tells the reader to go look at a Figure.
Line 231. Use reasonable significant figures, present integers for OTUs (1342 bacteria).
Line 233. Revise to “reached asymptotes (Fig. S1).”
Line 258. Revise to “demonstrated the potential to spoil grapes.”
Line 259. Revise to “..all of the microorganisms, except B. amyloliquefaciens, caused…”
Line 261. As above, revise to “all of the fungi, except…”
Line 272. Revise to “P. georgiense showed 100% spoilage on the McKinney index.”
Line 274. Again with the significant figures. Show integers here (52 not 51.57).
Line 294. Delete “the”
Line 297-299. Revise to “…with previous studies (Hall et al. 2019).” Delete “found that… throughput sequencing.”
Line 311. Avoid the word “believed” here. They reported.
Line 313. Delete “present among the”
Line 314. Delete “regarded as”
Line 317. Delete “been”
Line 320. The concept of antibiotic resistance comes out of nowhere.
Line 331. Present published facts in the present tense “is found” not “was found”
Line 332-353. I am not bothering to review this section. Follow the above examples.
Line 259. Delete “The diversity …Table 4.”
Line 354-362. Delete “Calvo…treatment of plant diseases.” You present no data on how to control grape rot. Biocontrol and storage methods are beyond the scope of this work.
Line 411. Here and throughout, present only volume and page numbers, not the issue (108:1429-1442).
Line 414. Is it “PLoS One” or “PLoS ONE (line 421)?” (the correct form is PLOS ONE)

---

## Round 0.6 · Minor Revisions

The manuscript appears close to acceptable and will not require another review, if you the comments are addressed adequately.

Michael

·

Basic reporting

Line 29: "Microorganisms causing sour rot": this should be changed to associated with sour rot

Line 49-51: As per Hall et al. 2018, sour rot also requires the presence of Drosophila fruit flies, and is also characterized by the smell of acetic acid.

Line 255: "Potential to spoil grapes" is different than causing sour rot symptoms. Sour rot symptoms are characterized by

Experimental design

No comment

Validity of the findings

Line 281: Grape sour rot has been shown to require the presence of fruit flies, in addition to yeast and acetic acid bacteria. Without fruit flies, necrosis occurs but the symptoms do not resemble sour rot in the field (Hall et al. 2018). The organisms associated with sour rot in your study are causing necrosis, but necrosis alone does not define sour rot. Please elaborate in your discussion.

Line 341 is the only mention that Drosophila are required for the disease symptoms to develop but even then, there is no discussion that Drosophila were not used in your assays. Please elaborate.

Additional comments

As the role of Drosophila fruit flies has been documented as a causal organism in sour rot development, I believe it needs to be mentioned in this paper. Also, I believe the lack of inclusion of fruit flies in the pathogenicity assays needs to be addressed.

---

## Round 0.7 · Minor Revisions

The manuscript still needs revision. Only my specific comments were addressed but those were just examples, not a comprehensive list. It is inappropriate for editors to ghost write manuscripts but I find myself rewriting this, which is an iterative process. Importantly, Discussion needs revision. I suggest these paragraphs in Discussion.

1. Start Discussion with a statement that follows the justification of the study provided in Abstract (line 27). I suggest “The microbes responsible for sour grape rot in eastern coastal China appear similar (or different) to those associated with this disease in vineyards around the world, as assessed by culture-dependent and metagenomic analysis.” Clarify that you only ID’d microbes that correlate with this disease not necessarily cause it.

2. Compare the bacteria you isolated to bacteria other researchers isolated from similar systems and you detected with metagenomics.

3. Compare the fungi you isolated to fungi other researchers isolated from similar systems and you detected with metagenomics.

4. Write a concluding paragraph that summarizes what you found out and why that matters, not what you didn’t do (look at fruit flies).

Specific comments.

Line 30. Revise to “…assessed by a culture-dependent techniques and metagenomic….”
Line 32. Replace “culturable microorganisms” with “isolates”
Line 35. Culturable and isolated is redundant. Replace “We isolated 15 culturable bacterial species and 10 fungal species from sour” with “We isolated 15 bacteria and 10 fungi from…”
Line 39. You need to make clear you are now present metagenomic data. Replace “sequencing revealed” with “sequencing of metagenomic DNA recovered from diseased grapes revealed”
Line 44. If Abstract has sections Background, Methods & Results, it must have sections “Discussion” and “Conclusions.” Discussion starts with “This suggests…”but as written it adds little. I suggest you discuss if the 15 microbes isolated are representative of the microbes that metagenomics suggested are associated with sour rot. I’d also discuss if the microbes that appear responsible for sour rot in eastern coastal areas of China are the same as those that found in other places.
Line 54. Replace “Due to the smell of acetic acid generated by microorganisms, Drosophila fruit flies are also contributed to cause grape sour rot...” with “Acetic acid released by AAB attracts the fruit fly Drosophila, which contributes to sour rot…”
Line 75. Replace “…2009). Great progress has been made in characterizing microbial diversity associated..” with “…2009) and characterization of microbes associated..”
Line 81. Start a new paragraph with “In this study…”
Line 82. Delete “a” in “and a”
Line 158. Provide references for the metagenomic sequence analysis pipeline (Mothur, QIME..). Also, deposit reads in the short read archive and provide a Bioproject number.
Line 206. The text “All bacterial taxa were Gram-negative confuses me.” Gram is a surname. Firmicutes are generally considered Gram positives. Do you mean all the Proteobacteria looked like Gram negatives?
Line 210. Replace “fungal species” with “fungi” and “belonged to” with “classified as”
Line 212. Put commas around “with bottle green colonies” and around “with a white …(line 213),” “with white ...(line 215) and similar phrases (line 216).
Line 223-225. Do not send the reader away to look at Tables and Figures. Replace “Sequences of these isolates showed > 97% similarity with those deposited in NCBI.... Due to high..” with “Sequences of these isolates showed high similarity to sequences of type strains (Fig. 2).” But, I am not sure what Figure 2 shows. You need to make clear which strains are from the study, by providing a strain designation, in which sequences come from reference sequences in Genbank.
Line 225-227. Make clear that you are discussing bacteria when you mention 16S rRNA genes and ITS for fungi. Fungi have 18S rRNA. Discuss bacterial and fungal phylogeny separately.
Line 230. Revise to “...gene of fungi generated from metagenomic DNA extracted from rotting grapes...”
Line 233. Replace “which indicated that the data quantity was sufficient” with “which suggests coverage was sufficient”
Line 248. Present reasonable significant figures (1% not 1.00%)
Line 432. Do not include issue (119(9):784-90) in references, just volume:page. See also 435, 439...
Give each isolate a strain identifier in Tables and Figures.
Table 4. Is it ± SD or SE? This Table is strange. Shannon values have no variation but ACE and Choi indices have variation greater than the mean.
Figure 2. The numbers on the branches are not defined.
Figure 4 should be about Fungi but it includes this “(A)The bacterial community...”

---

## Round 0.8 · Minor Revisions

I still have many edits.

Line 24. Replace
“...site. It has not been...the diversity and abundance” with “...site but the microbes responsible for this disease in eastern coastal China, has not been reported. To identify the microbes that cause sour grape rot in this important grape-producing region, the diversity and abundance...”

Line 30. Delete “further”

Line 31. Here and throughout, do not use the term “species” to describe the microbes you isolated. You generated a library of isolates that were subsequently classified as species or strains. Replace “bacterial species” with “bacteria” and replace “fungal species” with “fungi”

Line 41. Delete “multiple” and “isolated are”

Line 43. Replace “found in” with “responsible for this disease” and delete “as assessed...analysis.”

Line 49. Replace “It” with “This disease”

Line 97. Replace “inoculated in” with “spread on”

Line 98. I thought you inoculated nutrient agar not broth and we need to know the number of colonies picked. Replace “After culturing... using the primary medium.” With “After incubating at 25 °C for 48 h on nutrient agar and 7 d on potato dextrose agar a total of ____ colonies were picked and restreaked for purity using the primary media.”

Line 154. Deposit MiSeq reads in the short read archive and provide accession numbers here.

Line 200. Revise Tables 1 and 2 to include a column for the strain IDs. For example, Cronobacter sakazkii is strain SRG10. Then use these strain IDs (C. sakazkii SRG10) when discussing these strains, so the reader knows you are talking about a strain you isolated but not the species in general. Change Table 1 title to. “Phylogeny of microbes isolated from sour rotted grapes”

Line 206. Replace “...negative. Cronobacter...” with “Strains that classified as Cronobacter...”

Line 213. Separate phrases, like “with white colonies,” with commas and, “with red colonies.”

Line 221. Start paragraphs with a topic sentence that presents something of interest to microbiologists, not instructions to go look at data somewhere else. Delete “Sequences...(Table 1).”

Line 222. We cannot publish a tree that has the genera Bacillus appear on three different branches, with two of those branches including Gram negatives. Bacillus sp 1 and sp 2 should be on the same branch as Bacillus amyloliquefaciens. Also, how is Klebsiella more related to Lactobacillus than anything else? Make a new tree that has includes type strains that shows a reasonable topology. I don’t know much about fungi, you’d think fungal genera would share a branch too.

Line 283-290. Start Discussion with a topic sentence that relates to the specific goal of the study, which was to determine if the microbes that cause sour rot in China are different than the microbes that cause this disease everywhere else. Delete “As a serious...61% of the bacteria.”

Line 298. Start paragraphs talking about your results, not with the surname of another author “Barata..” Start of the paragraph with “We identified several Aspergillus strains associated with sour rot of grapes in China, which agrees with previous reports that molds contribute to sour-rot in grapes (refs).”
Line 302. Replace “The fungi excluding yeasts” with “Molds”

Line 336. Delete “active”

Line 337. Revise to “biological control agents”

Line 346. Again, don’t start a paragraph with another scientist’s name, unless it is Einstein, Newton... Make the subject of your topic sentences microbes not men.

Line 350. Do not use Discussion to call for “further study”

Line 364. Revise to “different than those”

Line 365. What are you contrasting with “however.” Replace “However, it was firstly proved that some species, such as A...” with “We report here that A... can spoil grapes.”

Line 368. Delete “Moreover..in our future study.”

---

## Round 0.9 · Minor Revisions

I still don’t know what the main conclusion of this work is. Are the microbes that spoil grapes in China different than those that spoil grapes around the world? Select one option: “similar to” or “different from”. The submitted text says they are different but then you report that many of the microbes you observed or isolated have been previously associated with rotting grapes (line 277). Also. Stick to one order throughout. Metagenomic then isolation. Bacteria then fungi.
Line 23 -42. Try this abstract, but note you have to select either “similar to” or “different from”
As a polymicrobial disease, sour rot decreases grape berry yield and wine quality. The diversity of microbial communities in sour rot-affected grapes depends on the cultivation site, but the microbes responsible for this disease in eastern coastal China, has not been reported. To identify the microbes that cause sour grape rot in this important grape-producing region, the diversity and abundance of bacteria and fungi were assessed by metagenomic analysis of 16S rDNA and ITS genes and 15 bacteria and 10 fungi were isolated from sour rot-affected grapes. High-throughput sequencing of PCR-amplicons generated from diseased grapes revealed 1343 OTUs of bacteria and 1038 OTUs of fungi. Proteobacteria and Firmicutes were dominant phyla among the 19 bacterial phyla identified. Ascomycota was the dominant fungal phylum and the fungi Issatchenkia terricola, Colletotrichum viniferum, Hanseniaspora vineae, Saprochaete gigas, and Candida diversa represented the vast majority of microbial species associated with sour rot-affected grapes. An in vitro spoilage assay confirmed that four of the isolated bacteria strains (two Cronobacter species, Serratia marcescens and Lysinibacillus fusiformis) and five of the isolated fungi strains (three Aspergillus species, Alternaria tenuissima, and Fusarium proliferatum) could spoil grapes. The microbes responsible for sour rot of grapes and the microbes responsible for sour rot in eastern coastal areas of China were similar to/different from those that responsible for this disease in vineyards around the world.
Line 63. Here and throughout reference citations as Author et al. (year) if there are more than two authors.
Line 77-81. Replace with and select either similar to or different from.
In this study, metagenomic analysis was used to determine the diversity and abundance of bacteria and fungi in sour rot-affected table grapes collected from Yantai city. In parallel, we isolated several microbes and determined their potential to spoil grapes in an in vitro assay. The microbes associated with this plant disease were similar to/different from those reported from other regions.

Line 201. Here and throughout, separate phrase with a comma (…Proteobacter, which were…).
Line 203. Use parallel sentence structures. If you say “presented positive results in …” then you should say “and presented negative results for fermentation with the methyl red..”
Line 205. As above, revise to “…pneumonia presented negative results for these biochemical..”
Line 276-290. Replace “In this study…and California.” With:
Metagenomic analysis and culturing indicated that the microbes that cause sour rotting of grapes in similar (I think) around the world. AAB were the dominant bacteria in rot-affected grapes in eastern coastal areas of China, which is consistent with reports from Australia (Mateo et al. (2014), Portugal (Baratta et al. 2012b) and New York (Hall et al. 2019). Aspergillis were the dominant mold, which agrees with reports from Greece (Tjamos et al., 2004) and California (Rooney-Latham et al., 2008). I. terricola, H. vineae, and C. diversa, Colletotrichum viniferum, and Saprochaete gigas, were also commonly observed, which are the same genera, but different species, than reports from other regions (Guerzoni & Marchetti, 1987; Barata et al. 2008; Barata et al. 2012b; Lleixà et al., 2018).

Line 293. Revise to “…California. We observed similar communities and report the presence of pathogens, that can infect humans and animals, associated with rotting grapes, such as..”

Line 302-308. I moved this section, so delete “Issatchenkia … of China.”

Line 309. Greater than what?

Line 344. I think there are more similarities than differences in the microbes you observed to the microbes others have reported.

Line 387. Here and throughout. The format for reference is volume:page-page (53:571-576). Do not provide the issue.

Table 2. The last six isolates are missing their strain IDs

Table 3. Use consistent and reasonable significant figures. Present all mean lengths of reads as whole numbers (449 not 448.9).

---

## Round 0.10 · Minor Revisions

The reviewers did not ask for more experiments, so I consider the changes minor.

Reviewer 1 ·

Basic reporting

1 BASIC REPORTING
Clear, unambiguous, professional English language used throughout.
The English language needs improvement. In particular substantial review for the style is necessary.
Intro & background to show context. Literature well referenced & relevant.
The introduction section has been improved. Please revise according to specific comments below.
Structure conforms to PeerJ Standards, discipline norm, or improved for clarity.
The structure conforms to PeerJ Standards.
Figures are relevant, high quality, well labelled & described.
Figures are relevant, of good quality, labelled and described as shown by figure captions and in the main text. Please revise according to specific comments below.
Raw data supplied.
Raw data are supplied in the supplementary files.

Experimental design

2 EXPERIMENTAL DESIGN
Original primary research within Scope of the journal.
Research is in accordance with Scope of the journal.
Research question well defined, relevant & meaningful. It is stated how the research fills and identified knowledge gap.
Research objectives are well defined, relevant, and meaningful:
The workflow has been revised.
Rigorous investigation performed to a high technical & ethical standard.
Research activities are performed following high technical and ethical standard.
Methods described with sufficient detail & information to replicate.
The methods are described with sufficient information regarding procedures and materials used.

Validity of the findings

3 VALIDITY OF THE FINDINGS
Impact and novelty not assessed. Negative/inconclusive results accepted. Meaningful replication encouraged where rationale & benefit to literature is clearly stated.
All underlying data have been provided; they are robust, statistically sound & controlled.
Data have been provided and they are relevant, statistically analysed and checked. Please revise according to specific comments below
Conclusions are well stated, linked to original research question & limited to supporting results.
Conclusions have been revised but still need minor corrections. See below.
Speculation is welcome, but should be identified as such.
-

Additional comments

Dear authors,
the manuscript has been carefully revised in its structure, changing the order of results presentation as correctly suggested by the Editor. However, this Reviewer still detected minor changes to be made throughout the manuscript. In particular, several errors highlighted by the Editor in the previous revision were not corrected. In addition, other errors and style mistakes need a substantial correction. For these reason I strongly suggest a "Final Minor Revision" prior to the publication in Peer J.

Please revise according to the following specific comments:
Specific comments
Line 44. Italicize “in vitro” and the microbial species (two Cronobacter species, Serratia marcescens and Lysinibacillus fusiformis and so on). Carefully check throughout the manuscript. Italicize also phylum, class, microbial family and microbial genera!!
L48 Delete could. These species spoil grapes as revealed by the spoilage in vitro assay!!
Lines 48-51 Please revise in: This suggests that the microorganisms responsible for grape spoilage in eastern coastal areas of China are in accordance with those isolated in vineyards worldwide. For the first time, opportunistic pathogens were isolated from rotten berries.
L73-74 Please use et al. when the authors are more than two. Please check the references throughout the manuscript
L79 Add the sentence: This microbial consortium produces spoilage metabolites such as acetic acid and gluconic acid (Pinto et al., 2019).
Add this reference (Pinto L., Malfeito-Ferreira M., Quintieri L., Silva A. C., Baruzzi F. 2019. Growth and metabolite production of a grape sour rot yeast-bacterium consortium on different carbon sources. International journal of food microbiology, 296: 65-74. DOI: 10.1016/j.ijfoodmicro.2019.02.022) in the References
Line 70-71. Avoid passive voice revise to “Basidomycetes.. dominate intact grape”
L85 Replace with …than in asymptomatic ones.
L140 Mothur with capital letter. Please revise
L287-291 Please revise in: “We identified Firmicutes as the dominant phylum (60%), with nine Gram-positive bacteria species (i.e., Staphylococcus saprophyticus, Lactococcus garvieae, Lactobacillus plantarum, two Lysinibacillus species, and four Bacillus species). We also isolated six Gram-negative bacteria species assigned to the phylum Proteobacteria.”
L292 Remove ; and replace with whereas
L293 Remove that
L343-344 Replace sour rot with spoilage
L385 Replace “microbes” with “microorganisms” and “sour rotting of grapes in” with “spoilage is”
L430 Correct with “the biofilm formation…”
L439 ethanol to acetic acid (Pinto et al., 2019). Please add this reference already suggested for the inclusion in the Introduction section.
L454-455 Revise in: “The microbes associated with sour grape rot in eastern coastal China appear similar to those associated with this disease…”
L457 Delete can.
Table 1 Delete means
Table 2 Present all parameters as integers. Replace “OTUs diversity, number” with “Diversity indices of”
Fig 2 Legend. Insert space in “ongenus”
Figure 3 legend. Replace “lables” with “labels”
Figure 5 Revise in: The percentage of incidence of microorganism using merged (A) or separated (C) methods; McKinney index of microorganism using merged (B) or separated (D) methods. Different letters in each figure indicate significant difference between microorganisms (One-way ANOVA; α = 0.05)

·

Basic reporting

Abstract: All scientific names but be italicized

Line 177-178: This scale does not establish the presence/absence of sour rot symptoms, but instead whether the berry was successfully colonized by the inoculated microorganism. “Disease spot” is not an accurate term in this context, as this test is not rating disease symptoms.

Line 245: While Figure 4A shows that the organisms have indeed colonized the berries, none of the berries show sour rot symptoms: lack of turgor in the berry, liquefaction of the berry pulp. All the berries appear intact, which is not consistent with sour rot. I recommend that the authors note this distinction – these organisms are associated with sour rot-affected grapes and not necessarily the causal organisms. If the authors want to see if these organisms can cause symptoms, they must include axenic Drosophila in the experiment, as fruit flies are causal organisms in this disease complex (Hall et al. 2018). What is the purpose of including the infection rates of these microorganisms if they are not causal? The authors may want to consider excluding these if they are not necessary, so as to make the paper clearer.

Citations: Formatting changes required throughout.

Tables 1, 2, 3 and 4 titles: Should be changed to ‘sour rot-affected grapes’

Table 3: Formatting needs to be corrected so words are not cut off on two lines

Figure 2: “on genus” and “on phylum” are two words each

Figures 3 and 4: All scientific names but be italicized

Experimental design

No comment

Validity of the findings

No comment

---

## Round 0.11 · Minor Revisions

I think the manuscript is much closer to acceptance and that I understand what you are reporting.

I have minor comments to the Abstract.
Line 27-28. Replace “by metagenomic analysis of 16S rDNA and ITS genes and 15 bacteria...” with “by metagenomic analysis and cultivation-dependent techniques. A total of 15 bacteria...”

Line 37-40. Replace “This suggests that the microorganisms ...isolated from rotten berries.” With “These microorganisms, which appear responsible for spoiling grapes in eastern China, appear closely related to microbes that cause this plant disease around the world.”

Line 39. As above. Delete “For the first time...berries.”

Line 410. Here and throughout. The format for reference is volume:page-page (53:571-576). Do not provide the issue. I have made this comment before and it infuriates me that it isn’t addressed. This is the author’s responsibility not the editors. Assign one co-author to carefully proof read every citation.

---

## Round 0.12 · accepted · Accept

There are some format error is References but those can be address in proof.

PLoS One (line 407) vs PLoS ONE (line 414)
Plant disease (line 418)
International journal of food microbiology (line 449)